# Interactions of Hematopoietic and Associated Mesenchymal Stem Cell Populations in the Bone Marrow Microenvironment, In Vivo and In Vitro Model

**DOI:** 10.3390/ijms26189036

**Published:** 2025-09-17

**Authors:** Darina Bačenková, Marianna Trebuňová, Erik Dosedla, Jana Čajková, Jozef Živčák

**Affiliations:** 1Department of Biomedical Engineering and Measurement, Faculty of Mechanical Engineering, Technical University of Košice, Letná 9, 042 00 Košice, Slovakia; marianna.trebunova@tuke.sk (M.T.); jana.cajkova@tuke.sk (J.Č.); jozef.zivcak@tuke.sk (J.Ž.); 2Department of Biomedical Research and Clinical Trials, L. Pasteur University Hospital in Košice, Rastislavova 43, 041 90 Košice, Slovakia; 3Department of Gynecology and Obstetrics, Faculty of Medicine, Pavol Jozef Šafarik Univerzity Hospital AGEL Košice-Šaca, Pavol Jozef Šafarik University in Kosice, 040 15 Košice-Šaca, Slovakia; erik.dosedla@nke.agel.sk

**Keywords:** hematopoietic stem cells, hematopoietic progenitor cells, mesenchymal stem cells, bone marrow microenvironment, bone marrow, collagens, fibronectin, hyaluronic acid

## Abstract

Multipotent hematopoietic stem cells (HSC) reside in specialized niches of the bone marrow (BM). The maintenance of their stemness requires a precisely regulated bone marrow microenvironment (BMM), supported by mesenchymal stem cells (MSCs), stromal reticular cells, and endothelial and nerve cells located within the vascular and endosteal niches. The heterogeneity of the niche environment is caused by the diversity of cell populations from HSCs to more mature hematopoietic cell types and MSCs, which collectively influence the complex intercellular interactions involved in hematopoiesis. MSC subclusters in BM are characterized by the phenotypes of CXC-chemokine ligand 12, leptin receptor, neuron-glial antigen 2, and Nestin+ cells. The article presents a detailed characterization of individual stem cell types in the BM, their reciprocal interaction, and the possibility of in vitro simulation of the bone marrow niche as a dynamic structure. Development of a suitable simulation of the BMM is essential for advancing research into both physiological and pathological processes of hematopoiesis. The main goal is to simulate 3D cell culture using biomaterials that mimic the BM niche in the form of hydrogels and scaffolds, in combination with extracellular matrix components.

## 1. Introduction

Interactions between bone marrow hematopoietic stem cells (HSCs) and non-hematopoietic cell populations within the bone marrow (BM) microenvironment are currently the subject of intensive research, both in vivo and in vitro, due to their fundamental role in regulating hematopoiesis [1,2,3]. In the BM niche, HSCs and hematopoetic progenitor cells (HPCs) interact with mesenchymal stem cells (MSCs) populations and differentiated cell types such as endosteal osteoblasts, leptin receptor (LepR)-positive cells, CXCL12-rich reticular cells (CAR) cells, and neuroglial antigen 2 (NG2+) cells [1]. Also, extracellular matrix (ECM) components such as collagens, proteoglycans, and glycoproteins play a role in processes in the BM niche. These cellular and non-cellular components shape the finely balanced BM niche, influencing HSC function and hematopoiesis—a continuous but highly adaptive process that responds to the organism’s physiological needs [2]. In this manuscript, we aim to contribute to the current understanding of the BM microenvironment by summarizing key data on its cellular interactions and the individual components of the ECM. At the end of the article, we discuss the possibility of creating models simulating the BM microenvironment in vitro. It has been postulated that the resulting differentiation ability of HSCs depends on the surrounding BM environment, which has been defined as the hematopoietic inductive microenvironment. The differentiation of HSCs is therefore stimulated by the environment of the stroma filling the BM [4]. In the BM, niche HSC cells are in direct contact with stromal cells or with the ECM to maintain multilineage potential “stemness”, self-renewal capacity, and differentiation into all hematopoietic cell lineages [5]. BM is a rich source of HSCs, which are responsible for the formation and maintenance of the cellular components of blood [6]. The presence of hematopoietic stem cells was demonstrated in an in vivo experiment in 1961 by Till and McCullough by monitoring the formation of spleen colonies (CFU-S). They exposed a group of laboratory mice to high doses of radiation, which had a strong cytotoxic effect on their hematopoietic cells. BM cells from related individuals were transplanted into the animals, thereby inducing the restoration of hematopoiesis [7]. Early in the 1970s, Schofield presented his concept of an ordered distribution in a “niche” of HSCs in the bone marrow microenvironment (BMM) [8]. Adams et al. proposed that BMM stimuli are involved in the regulation of hematopoietic stem cells through a highly specific environment. Several authors have addressed the topic of the HSC niche model [9]. The HSC niche is generally described as perisinusoidal, near the sinusoids, and the endosteum region mainly provides a niche for HSCs [10,11]. In the osteoblastic niche model, the stromal part is in contact with the endosteum, and the result of MSC differentiation can be a subtype of osteoblasts and stromal cell subtypes [12]. This confirmed the presence of a vacuolar niche, which stimulated the development of research on the BMM [2,13]. In the vascular environment model, the stromal component is the cell that forms the vascular sinus. Interestingly, HSCs are fully characterized by their immunophenotype, but their precise localization is difficult, mainly due to the properties of the so-called “soluble tissue” found in bone tissue [14].

## 2. Heterogeneous Populations of Bone Marrow Stem Cells

### 2.1. Hematopoietic Stem/Progenitor Cells

In adulthood, the production of blood cells of all types occurs exclusively through the HSCs of the BM. HSCs form the foundation of the hematopoietic system; however, recent evidence indicates that they constitute a heterogeneous rather than a homogeneous cell population [15]. The process of hematopoiesis is strictly regulated by stimulatory and inhibitory growth factors [16]. The expression of various receptors on the surface of hematopoietic progenitors allows interaction with regulatory elements present in the environment, which include stromal cells, ECM molecules, regulatory cytokines, growth, and differentiation factors. It is assumed that in the BM of an adult individual, only one in 10 to 15 thousand cells is a BM stem cell. Analyzing the localization of human hematopoietic stem and progenitor cells (HSCs/HPCs) is more challenging and is mainly performed microscopically through histological methods of fluorescently specifically labeled stem cell populations. From experiments on murine KB sections, the authors detected the presence of HSCs/HPCs in the perivascular niche, in the perisunosoidal region [17,18]. Studies have shown that HSCs/HPCs are located near sinusoids in the human BM [19], in the area near the endosteum, cancellous bone area. In murine experiments [20], HSCs are considered to be the apex of the hematopoietic hierarchy. They are currently defined as clonogenic cells capable of self-renewal and differentiation in the process of multilineage hematopoiesis [21]. HSCs are multipotent with the ability to differentiate into HPCs, which form the basis of hematopoietic lineages through lymphoid progenitors giving rise to the B cell, T cell, and natural killer (NK) cell population, and through myeloid progenitors, they differentiate into red blood cells, platelets, granulocytes, and monocytes. The heterogeneous nature of HSCs relates to several properties, including self-renewal, lifespan, and differentiation [22]. HPCs play a crucial role in maintaining homeostatic blood cell levels and in supporting proliferation during sudden fluctuations in cell numbers, including the activation of specific stem cell clones. During quiescence, the BM acts as a reservoir for resting HSCs [23].

#### HSC/HPC Phenotype

The CD mark panel characteristic of HSCs/HPCs is optimized through experimental data acquisition through animal experiments in murine and clinical data in humans. Identification of the phenotype of HSC and HPC is essential for the isolation and purification of these populations. Experimental work investigating animal models, mostly from the BM of adult mice, is essential for characterizing the phenotype of hematopoietic stem cells [24,25]. According to the degree of HSC lifespan, populations of long-term (LT-HSC) and short-term repopulating HSCs (ST-HSC) are described from experimental results in mice [26]. A panel of markers for the characterization of mouse HSCs was defined as CD150+ CD48− CD34lo/negCD117+ SCA1+ (Lin−) for the isolation of adult mouse HSCs with long-term self-renewal capacity [27].

The murine phenotype of hematopoietic lineages has been well characterized and is based on the selection of the phenotype of “Lin^−^ c-Kit1^+^ Sca1^+^” (LKS) cells by flow cytometry [28]. Murine LKS cells are responsible for a stable population of hematopoietic cells and have a high rate of regeneration.

Research and characterization of the phenotype of HSCs and HPCs in humans focuses on stem cells isolated from BM and, to a lesser extent, cord blood [22,29].

“Hematopoietic Progenitor Cell Antigen CD34” (CD34) has been described on human HSC/HPC cells [30,31,32,33]. The rare CD34-negative HSC population is hierarchically superior to CD34-positive cells in human cord blood, and CD34-positive hematopoietic stem and progenitor cells have a fundamental function in hematopoiesis [34]. Human HSCs/HPCs are characterized by high expression of CD34, moderate expression of Proto-oncogene c-KIT (c-kit) and THYmocyte differentiation antigen 1 (Thy-1, or CD90), and low or no expression of CD38, HLA-DR, and CD71 [35]. In addition to BM, these cells are also found in umbilical cord blood and peripheral blood [36].

In the BM, there is a population of stem cells characterized as long-term and multipotent progenitor cells [37,38]. The phenotype of the CD34+ CD38− Long-term culture-initiating cells (LTC-IC) population derived from human BM is that of early progenitor cells with repopulating potential, with the ability to survive in a quiescent state, without mitoses. The LTC-IC assay is based on the observation of BM “stromal” adherent cells with the ability to support the survival of early hematopoietic cells, but also the production of colony-forming cells (CFCs) by these early hematopoietic cells. Clonogenic cells initially present in a cell suspension are usually not able to survive for a period longer than three weeks, the production of clonogenic cells after five to eight weeks can be used to quantify the number of primitive LTC-IC present at the time of initiation of in vitro culture. This test is able to detect a rare population of early, primitive cells that have the long-term potential to colonize healthy human or mouse BM [39].

Notta et al. identified the Integrin Subunit Alpha 6 (ITGA6), or otherwise adhesion molecule CD49f (VLA-6), as a specific HSC marker. The CD49f+ cell subpopulation was capable of forming long-term multilineage engraftments, while loss of CD49f expression identified a HPC subpopulation [22]. The characterization of stem cells as lineage-negative Lin- refers to a premature population of hematopoietic cells that do not express the phenotype of differentiated cell types. The population of LT-HSCs with the phenotype Lin− CD34+ CD38− CD45RA− CD90+ CD49f+, precisely referred to as human long-term repopulating hematopoietic stem cells, is also accepted to define a subset of CD34+ cord blood cells [22]. Very early progenitor cells are enriched within the CD34+ CD38− HLA-DR− Thy+ or CD34+ population, which lacks all differentiation markers, the so-called “CD34+ Lin−” population [40]. Recent data suggest that the next hematopoietic cell in the lineage is HPCs, forming a population composed of heterogeneous cells that possess different specific properties while retaining plasticity. HPCs can be distinguished based on a characteristic phenotype, without CD49 expression, such as Lin− CD34+ CD38− CD45RA− CD90− CD49f− [40,41]. Other authors report two comparable phenotype expressions for the identification of human HSCs used after mobilization in peripheral blood: the phenotype Lin− CD34+ CD38− [42] and Lin− CD90+ CD45RA− CD71− [43,44,45].

Both human and mouse HSCs express high levels of permeability glycoprotein (P-glycoprotein), a membrane efflux pump known for mediating drug resistance in tumor cells. P-glycoprotein is involved in the elimination of toxins and xenobiotics from cells [46]. CD38 functions as a recentor, or enzyme. The CD38 receptor binds to CD31 on T lymphocytes. CD38 acts as an enzyme, catalyzing the synthesis of ADP ribose from NAD+. Regarding CD38 expression, human HSCs are thought to be CD38-, but more mature progenitor cells become CD38+. In contrast, the majority of mouse HSCs are CD34-CD38+ [47].

Another surface antigen that defines HPCs is Prominin-1 (AC133, CD133). AC133 is a transmembrane glycoprotein antigen that is selectively expressed on most CD34+ cells from human fetal liver, BM, and normal and mobilized peripheral hematopoietic cells (PB) [48]. It is not detectable on other hematopoietic cells and provides an alternative marker to CD34 for the selection and characterization of hematopoietic cells necessary for short- and long-term engraftment [49]. However, the presence of CD34+, AC133+, and the absence of CD38−, HLA-DR-, and “Lineage-associated” markers defines a cell population with the ability to restore hematopoiesis in patients, recipients after a high myeloablative dose of chemoradiotherapy [48]. The identification of HSC/HPC populations is currently being studied experimentally. G protein-coupled receptor class C group 5 member C (GPRC5C) has been identified as a receptor for human dormant HSCs. In vivo human studies have demonstrated the function of GPRC5C in the maintenance of dormancy and stemness [50].

Growth factors actively interact through target cell receptors and activate signaling cascades that influence cellular metabolism. In the BM environment, there are specific factors that are produced by cells and target HSC proliferation, mobilization, and quiescence. Under severe stress conditions such as radiation, chemotherapy, Stem cell factor-1 (SCF-1) is released in the BM or liver [51]. HSCs residing in the perisinusoidal space adhere to endothelial cells (ECs) on one side, while maintaining contact with parenchymal cells on the opposite side. The chemokine CXCL12 (above mentioned under the title SDF-1) has an activating effect on HSCs. There is crosstalk with MSCs via CXCL12 and C–X–C chemokine receptor type 4 (CXCR4) on CD34+ HSCs [52]. CXCL12 is a chemokine involved in the mobilization and homing of CD34+ HSCs in homeostasis and during wound healing processes, and participates in tissue repair [53]. CXCL12 levels stimulate the mobilization of HSCs and progenitor cells towards an increased CXCL12 gradient [54]. When CD34+ cells enter the periphery, they migrate towards the chemoattractant CXCL12 and bind via the CXCR4 receptor. This initiates a series of signaling processes that enhance cell migration and adhesion [55]. CXCL12 has an effect on increased secretion of the angiogenic factor Vascular endothelial growth factor (VEGF) and metalloproteinases [56] (Table 1 and Figure 1).

### 2.2. Subsection Bone-Marrow Derived Mesenchymal Stem Cells and Subtypes of MSCs

Two main types of niches have been identified and characterized in the BM. The area surrounding the blood vessels in the BM is characterized as the “vascular or perivascular niche”. The area near the bony structures in the BM is characterized as the “endosteal or osteoblastic niche” [57,58]. The niche region is home to specific interacting cell populations, the localization of which partially overlaps their location in the BM [12,18]. The perisinusoidal niche is characterized by the presence of a heterogeneous cell population including MSCs, perivascular stromal cells, ECs, macrophages, CAR cells, Nestin+, andNG2+ cells, and Schwann cells interacting with HSCs [1]. The endosteal niche is composed predominantly of a population of osteoblasts and a smaller number of osteoclasts [59]. The mesenchymal origin of BM stromal cells has been well established. MSCs are a population of stem cells with the ability to differentiate into differentiated non-hematopoietic cell types in the BM niche environment [60,61]. Bone-marrow-derived MSCs (BM-MSCs) are part of the supportive hematopoietic stroma and contribute to the BMM [62]. BM-MSCs are essential for the physiological condition of the BM niche and act on HSC attachment and adherence. The ability of BM-MSCs to mediate hematopoiesis through cell–cell contact with HSCs and through growth factors has also been confirmed [63]. In humans, multipotent mesenchymal stromal cells/mesenchymal stem cells are isolated from KD aspirate, adipose tissue, and other tissues and are monolayer-cultured in vitro with a characteristic spindle-shaped shape as adherent cells [61]. In vitro cultured MSCs must be adherent to the culture plastic surface and express a phenotype CD105+, CD73+, and CD90+, without expression of hematopoietic markers CD45−, CD34−, CD14−, or CD11b−, CD79α−, or CD19− and HLA-DR-, according to International Society for Cell Therapy (ISCT) criteria [61]. They must be capable of differentiating into the three lineages, osteoblasts, chondrocytes, and adipocytes under appropriate stimulation conditions in vitro [61,64]. MSCs are self-renewing stem cells with the above-described differentiation capabilities that express a panel of key markers CD10+, CD13+, CD29+, CD73+, CD90+, CD105+, CD271+, CD146+, STRO-1+, and SSEA4+, and markers include CD140b+, Human Epidermal Growth Factor Receptor 2 (HER-2)+, and frizzled-9+ (CD349) [65,66,67].

MSC subclusters localized in the BM co-create a microenvironment with a phenotype typical of their position in BMM. The MSC population in the perivascular vicinity is characterized by the phenotype of CXCL12, LepR, and NG2+, also known as Chondroitin sulfate proteoglycan 4 and Nestin+ cells [1,3]. The endosteal niche is mainly populated by populations of differentiated MSCs, osteogenic progenitors, including osteoblasts and osteoclasts [1]. Adult BM-MSC populations can be further categorized by phenotypes of Lin−, CD45−, CD271+, and CD140a−/low markers, with CD146+ expression distinguishing perisinusoidal from endosteal CD146−/lo MSCs [68]. The proliferative cluster with the CD26+ phenotype has the ability to differentiate into other subpopulations [69]. The Chemokine-Like Receptor 1 (CMKLR1)+ cluster cells had immunomodulatory potential and were capable of preferential osteogenic differentiation, but had lower proliferative capacity and reduced adipogenic differentiation [69,70] (Figure 2).

#### 2.2.1. Endosteal Osteoblasts–Osteolineage Cells

The transition, or interface, between bone tissue and BM is called the endosteum. Its surface is lined by osteoprogenitors, osteoblasts, and to a lesser extent, bone-resorbing osteoclasts. The BM environment is surrounded by bone tissue. Originally, osteoprogenitor cells differentiate from MSCs in the BM [12,71]. Osteogenic cells in the BM are a population of cells associated with the modulation of haematopoietic stem and progenitor cells (HSPCs) [72] and constitute the endosteal niche. Early HSCs are present in the BMM, primarily in the endosteal region of trabecular bone. In the endosteal niche, long-term and short-term HSCs are identifiable by phenotype [22]. Approximately 20–30% of quiescent HSCs are associated with endosteal osteoblasts [13]. Rapidly cycling HSCs are found around blood vessels in the subendosteal region [73]. The function of endosteal osteoblasts in the maintenance and self-renewal of HSCs was proposed by Taichman et al. The relationship between bone tissue and BM has been confirmed [74]. In animal models, a direct relationship between the number of osteoblasts and the number of long-term HSCs has been observed, in the sense that the quantitative number of both populations is maintained, which mutually influences each other [12]. This fact demonstrates the essential role of osteoblasts in HSCs. Osteolineage cells that are differentiated from the LepR+ MSC population have a phenotype with significant expression of alkaline phosphatase, osteopontin, and osteocalcin. Perivascular cells from genetically modified mice with a “Cre phenotype” that were differentiated from LepR, Transcription factor Sp7 (osterix), or Paired-related homeobox 1 (Prx-1)-cre have been confirmed to participate in HSC homeostasis [75,76]. Endosteal osteoblasts are described as Prx-1-Cre-derived cells with a reparative role during bone fracture healing [77]. Also, Runx2, expressed in early osteoblasts, marks a population that has the ability to maintain HSCs, suggesting a function of the immature osteoblastic population in regulating HSCs [78].

In bone tissue, the process of bone formation and resorption is constantly taking place through a population of osteoblasts and osteoclasts—cells involved in the remodeling of the surrounding tissue [79]. Osteoclasts, as cells present in the BM environment, may participate in the regulation of HSCs. They also play a role in the formation of sinusoids in the BM [80]. The application of Granulocyte colony-stimulating factor (G-CSF) has been shown to increase osteoclast activity in humans and laboratory rodents [81]. Osteoblasts expressing macrophage colony-stimulating factor (M-CSF) and receptor for nuclear factor-kappa B ligand (RANKL) are essential for osteoclast differentiation. These factors are expressed by stromal cells in the BM [82]. Osteoclasts mature through a multi-step process and only when fully differentiated when they are capable of resorbing bone tissue. Chemokines, such as chemokine (C–C motif) ligand 3 (CCL3a), are able to increase osteoclastogenesis. CXC chemokines can regulate angiogenesis, due to the NH2-terminal end of CXC chemokines containing a three-amino acid motif (GluLeu-Arg: ELR motif) that acts as a promoter of angiogenesis [83]. During the bone matrix degradation process, osteoclasts produce metalloproteinase 9 and cathepsin K, which proteolytically inactivates CXCL12, and thus HSCs are released from the BM. Grassi et al. demonstrated the expression of CXCL10 and CXCL12 in vitro [84]. Osteoblasts paracrinely influence the efficiency of osteoclast degradation of bone matrix [85].

#### 2.2.2. Leptin Receptor+ (LepR) Cells/Leptin+ MSC Osteoprogenitore

A substantial subset of MSCs expresses LepR, with some of these cells also co-expressing CXCL12 [86,87,88]. A population of LepR+ MSC osteoprogenitors that produce collagen type I (COL1) is found in the surrounding area of periarteriolar and trabecular bone tissue [89]. Several studies have shown a significant influence of osteogenic progenitors on HSCs in the BM. HSPCs are distributed preferentially along the endosteal region [90]. Their interaction with osteoblasts with the N-cadherin+ CD45- osteoblastic phenotype is important for the quiescence of HSCs residing in the endosteum. Interaction with Ang-1 and Ang receptor-2 is hypothesized [91]. Neural-cadherin (N-cadherin/CD325/CH2) is a transmembrane protein responsible for cell–cell adhesion and is an integral part of adherent junctions [92]. These areas are composed of osteoblasts, CXCL12-expressing reticular cells, nestin-positive mesenchymal cells, Schwann cells, and perivascular cells [57,73,93,94,95]. Nilsson et al. have shown that osteopontin, a protein overexpressed in osteoblasts, is a key molecule in the maintenance, regulation, proliferation, and localization of HSCs in the BM [96]. LepR+ cells express and produce cytokines with an impact on hematopoiesis. In the BM, LepR+ cells near sinusoids express higher SCF levels than periarteriolar LepR+ cells. LepR+ cells express SCF and CXCL12, which positively influence hematopoiesis, assisting with homing and maintenance. HSCs are influenced by SCF, which is produced by LepR+ cells and also by sinusoidal endothelial cells, while MPP is mainly influenced by SCF expressed by LepR+ cells [88].

#### 2.2.3. CXCL12 Abundant Reticular Cells (CAR)

The heterogeneity of the niche environment is shaped by the diverse cellular composition, ranging from HSCs to more differentiated hematopoietic cell types and MSCs, which collectively influence the complex intercellular interactions governing hematopoiesis. HSCs are predominantly localized in the sinusoids of the BM [73]. Perivascular stromal cells provide a niche for HSCs [90]. Numerous cell types of sinusoids and arterioles are involved in niche metabolism, namely, endothelial, perivascular CAR-positive, and nestin+ or leptin-receptor+ mesenchymal stromal cells [13,97,98]. A population of heterogeneous perivascular cell types cooperate to influence HPSCs, specifically CAR cells, Nestin+ MSCs, and CXCL12, or otherwise SDF-1 reticular cells [73].

CAR cells are differentiated from LepR+ cells, originally from MSCs. CAR cells are therefore characterized by CXCL12 production in the BM in the sinus region [99,100]. MSCs that are localized around sinusoids and arterioles influenced by adipocytes produce HSC-modulating factors, namely, CXCL12, or another SCF-1, interleukins (IL), and Bone morphogenetic protein (BMP) 4 [101]. HSCs are believed to interact with MSCs through CXCL12–CXCR4 signaling, and with osteochondrogenic progenitor cells via secreted phosphoprotein 1 (SPP1)–CD44 crosstalk [58]. Analysis of BM samples identified that SPP1-expressing OCs were localized near the endosteal region and that CD271-positive stromal cells were found in the perivascular and stromal regions, suggesting variation in the stromal cell population within specialized niches for hematopoietic cells [102]. CXCL12 has the ability to bind through the HSC receptor CXCR4. Thus, HSCs are stimulated in the BM by CXCL12-CXCR4 binding to LepR+ CAR cells, a process that contributes to CXCL12 accumulation in the BM and acts as a chemoattractant gradient for HSCs [103].

Several authors have observed a significant role for CAR cells in HSC function in mice [87]. Although similarities between the mouse and human niches are assumed, there are still open questions regarding the details of its function. This includes the spatial localization and abundance of HSCs within the niche, as well as the identification of specific cell types essential for maintaining stem cell niche function [104]. Another relevant population is Nestin-GFP+ perivascular stromal cells, whose phenotype overlaps with CAR cells as stromal cells, which are capable of differentiation into adipocyte and osteoblastic cell types [105].

#### 2.2.4. Nestin and Neuron-Glial Antigen 2 Cells (NG2)

Nestin^+^ MSCs co-expressing neuron-glial antigen 2 are naturally localized around periarteriolar niches and exhibit little-to-no expression of CXCL12 and SCF. Their depletion has only a minimal impact on HSC numbers [76,93]. A hypothesis has been proposed for a supporting function of a portion of MSCs in early hematopoiesis in the BM, with expression of Ang-1, osteopontin, IL-7, and vascular cell adhesion molecule 1 (VCAM-1) being reported [87]. Hematopoiesis in the BM is strictly regulated by the nervous system. Autonomic nerves extend into the BM, reaching regions of active hematopoiesis, which exhibit the highest density of innervation [106]. They maintain the homeostasis of the HSC niche in a state of equilibrium. The BM and bone tissue environment is innervated by sympathetic and sensory nerve cells, which are essential for the process of hematopoiesis. Peripheral sympathetic neurons and Schwann cells are a significant part of the BM niche [107]. Circadian secretion of noradrenaline by sympathetic nerve terminals influences circadian expression of CXCL12, Nestin+/NG2+ perivascular MSCs, which in turn influences the controlled, cyclic release of HSCs into the periphery. Neurogenic signals are mediated via adrenergic receptors. The secretion of HSC-regulatory factors, such as SCF and CXCL12, by Nestin^+^/NG2^+^ MSCs supports the maintenance of HSC quiescence in regions associated with arterioles innervated by nerve fibers. Activated HSCs are attracted to Nestin—the leptin receptor in the perisinusoidal region. Schwann cells act on resting-stage HSCs through transforming growth factor β (TGF-β)/SMAD signaling [107].

#### 2.2.5. Pericytes

Recent studies indicate that the origin of BM-derived vascular cells extends beyond ECs to perivascular cells. Anatomically, pericytes surround the surface layer of capillary endothelium and are in close contact with the capillaries. They are located in the basement membrane space, where communication with the vascular wall and paracrine signaling occur [108]. The presence of pericyte progenitor cells together with EPCs may aid in vasculature remodeling and maintenance [109]. The localization of pericytes on microcapillaries is not random, but is functionally determined. The interaction between pericytes and vascular endothelium is important for the maturation, remodeling, and maintenance of the vascular system through the secretion of growth factors or the formation of ECM [110]. Morphologically, pericytes have an elongated, stellate (star-shaped) morphology with contact with the surface of ECs. The pericyte cell body consists of a prominent nucleus with inconspicuous perinuclear cytoplasm. At the point of contact, there are mutual communication connections in the form of intercellular junctions, “gap junctions”, which connect the cytoplasms of both cell types and allow ion exchange via N-cadherin and connexin. Pericytes show structural and functional heterogeneity. The contact of ECs with pericytes differs significantly from the type of microcapillaries. Pericytes have a phenotype with similar characteristics to MSCs, expressing characteristic pericyte markers CD146+, CD34−, CD45−, and CD56− with the ability of subsequent in vitro expansion [111]. Pericytes are multipotent cells capable of differentiation into adipocytes, osteoblasts, and phagocytic cells. Pericytes have also been observed forming in vitro colonies containing calcium phosphate and ECM with the presence of alkaline phosphatase and COL [108]. Pericytes, locally located regulatory cells, are important for the maintenance of homeostasis and hemostasis. They are a source of adult pluripotent cells [112]. Pericytes contribute to the formation, maturation, and homeostasis of vascularized tissues [108,113].

#### 2.2.6. Endothelial Cells (ECs)

The hematopoietic BMM and the ECs of the BM, Bone marrow endothelial cells (BMEC), which form a barrier between the peripheral environment and the BM parenchyma, play an important role in the process of hematopoiesis. The BMM is populated by EC lining the interior of blood vessels. ECs express a characteristic Notch+, CXCL12+, SCF+, Vascular endothelial growth factor receptor (VEGFR)2, and pleiotrophin+ phenotype that influences the metabolism and cell cycles of HSCs and HPCs [57,87]. When compared to MSC cell types, the expression of SCF and CXCL12 is significantly lower [105]. The role of pleitropin, a heparin-binding protein (PTN) secreted by ECs, is mediated by the surrounding vascular microenvironment and acts on HSC self-renewal and retention [114]. ECs co-form the vascular niche for HSCs and LT-HSCs through the action of activation factors. In the absence of these factors, the balance of HSC/HPC function is disrupted. Studies have shown that blocking ECs with specific antibodies reduced HSC engraftment in vivo [115]. ECs are divided into arteriolar ECs (AECs) and sinusoidal ECs (SECs) based on their location [87,116]. Using CD150, CD48, and CD41 antibodies to label endogenous HSCs, their localization was identified in close proximity to sinusoidal ECs [93]. SCF is likely expressed by BM fibroblasts, osteoblasts, and perivascular MSCs with CXCL12 phenotype and Nestin-positive cells in addition to ECs [117]. ACFs in the endothelium strongly express SCF and express the glycoprotein developmental endothelial locus (DEL1), which stimulates HSC proliferation and myeloid lineage progression [118] (Table 2).

### 2.3. Mobilization and Homing of Stem Cells

Stem cell homing is a physiological process that occurs endogenously, while administered therapeutic HSCs/HPCs utilize the same mechanism. This involves a multistep process that is critical to the success of stem cell transplantation. Modulating this process can enhance engraftment efficiency and provide deeper insight into the underlying mechanisms of stem cell migration [119]. The main role of stem cells occurring in the peripheral circulation of adult organisms is still a matter of debate [120]. The success of BM transplantation via intravenous infusion relies on the ability of HSCs/HPCs to “home” and engraft within the recipient’s BM. To mobilize progenitor cells, patients are given G-CSF. Homing is a complex process that begins with specific molecular intercellular recognition, adhesion and release, transendothelial migration, and functional repopulation of the depleted BM environment with stem cells [121]. For efficient homing of circulating HSC/HPC and BM colonization, high concentrations of CXCL12, CXCR4, and the adhesion molecules Very late antigen-4 (VLA-4) and Lymphocyte function-associated antigen 1 (LFA-1) are required [121].

Mobilized progenitor cells currently represent the most commonly used source of HPCs to perform hematopoietic reconstitution after myeloablative chemotherapies. The results of the Levesque authors point to a role for the adhesion molecule VCAM-1 (CD106) in the BM during HPC mobilization. VCAM-1 expression is downregulated in vivo in the BM upon G-CSF stimulation. The process is thought to involve serine proteases, specifically neutrophil elastase and cathepsin G, which cleave VCAM-1. The proteases are produced by neutrophils in the BM under the influence of G-CSF [122]. Recently, a VCAM-1+ macrophage-like cell population was identified that interacts with HSCs in an ITGA4-dependent manner and influences HSPC retention. The cells were characterized as “homing cells,” with a significant role in the homing microenvironment [123].

**Table 2 ijms-26-09036-t002:** Characteristics of non-hematopoietic cell populations in the bone marrow in the endosteal niche, periarteriolar niche, and sinusoidal endothelial niche. Abbreviations. Alkaline phosphatase (ALP), Angiopoietin-1 (Ang-1), Bone morphogenetic protein 4 (BMP 4), CXCL12 abundant reticular cells (CAR) cells, C–X–C chemokine receptor type 4 (CXCR4), hematopoietic stem cells (HSCs), mesenchymal stem cells (MSCs), Neuron-glial antigen 2 (NG2+), Type I collagen (COL1), Proteins platelet-derived growth factor receptor-alpha (PDGFR-α), leptin receptor (LepR), Stromal cell surface marker 1 (STRO-1), Vascular endothelial growth factor receptor (VEGFR), Vascular cell adhesion molecule 1 (VCAM-1).

Type of Niche Bone Marrow	Cell Type	Cell Phenotype	HSC Assignment	References
Endosteal niche	Osteoblasts	ALP, COL1, osteopontin	Influence quiescence of HSCs Homing for exogenous HSCs	[1,121]
Periarteriolar niche	Nestin+ cells perivascular MSCs	Nestin, PDGFR-α	Circadian oscillations of HSC release, HSC homing	[93]
	NG2+ cells perivascular MSCs	NG 2 Ang-1, VCAM-1	HSC maintenance and activation	[87,123]
Sinusoidal Endothelial/Perisinusoidal niche	CAR cells	CXCL12, BMP 4	HSCs of interaction with CAR cells via CXCL12-CXCR4	[58,99]
	LepR+ cells	LepR co-expresion CXCL12	Influence quiescence of HSCs	[87]
	Nestin+ cells	Nestin, PDGFR-alpha	HSC homing	[93]
	Endothelial cells	Stro-1, VEGFR2 Notch, CXCL12	The surrounding vascular microenvironment and acts on HSC self-renewal and retention	[1,114]

## 3. The Extracellular Matrix of Hematopoietic Stem Cell Niches

The ECM mediates all aspects of the signaling pathways involved in the biological processes of HSCs. Niche cells constitute a specialized microenvironment that maintains the pluripotency, “stemness”, and progenitor characteristics of HSCs, and are situated in their immediate vicinity [13]. The bone marrow stroma is formed by small arterioles with sinusoidal capillaries, forming a three-dimensional network of reticular cells with the ability to phagocytose and fine mesh reticular fibers, which contain stem and hematopoietic cells. It creates the so-called hematological niche. The BM matrix consists of multiple types of collagens (COLs), fibronectin (FN), laminin, and proteoglycans. Laminin and FN together form an environment for cell progenitors, which are connected in the ECM environment through receptors [124]. HSCs interact through cell receptors. Cell adhesion molecules (CAMs) are the most common elements that mediate cell–matrix interactions. They also function in signal transduction and are capable of acting as mechanoreceptors of the microenvironment [125]. The main types of CAM families include the cadherin family, the selectin family, the immunoglobulin superfamily, and the integrin family. CAMs, in addition to interacting with each other, allow adhesion to ECM molecules [126]. ECs and adipocytes are involved in the synthesis of basement membrane structures. HSCs and hematopoietic progenitors respond to the presence of biomechanical signals, and their incorporation of ECs into the niche ultimately influences hematopoietic cell differentiation [127]. The ECM naturally binds growth factors and makes them structurally accessible for intercellular interactions. Specialized BM cells produce active molecules that mediate HSC activity. A key one is Stromal cell-derived factor 1, also known as CXCL-12, Ang-1, or SDF-1 [128]. The role of factors acting in the BM is currently being intensively studied, along with the interaction of ECM components. ECM molecules are the main components responsible for the properties of the BM niche [129].

### 3.1. Glycoproteins

Glycoproteins, a family of glycosylated proteins with a broad range of functions in the BM, are essential for the construction of the hematopoietic niche. The laminin family, which belongs to the integrins, consists of large heterotrimeric molecules of alpha, beta, and gamma chains. They have diverse biological functions. Laminins are found in the basement membrane underlying epithelial and ECs [130]. The alpha4 and alpha5 chain laminin isoforms are found in the BM [131]. Experimental results have shown that the alpha5 chain acts as an adhesion molecule for erythroid cell lines [132]. Netrins are proteins that associate with laminin molecules in the processes of adhesion, proliferation, migration, and differentiation. Netrin-1 influences HSC self-renewal and quiescence are influenced by factors produced by osteoclast precursors, which exert paracrine effects on their maturation into more differentiated osteoclastic stages [133].

Netrin-1 binding is mediated through the cell surface receptor neogenin-1, which is expressed on a subset of dormant HSCs. Interestingly, a decrease in netrin-1 expression has been observed during the aging process. The decrease in netrin-1 synthesis is compensated by the upregulation of neogenin-1 in HSCs of the aging organism [134,135]. In the BM, Netrin-4 is produced by endothelial and perivascular cells, and can act as an inhibitor of osteoclast maturation [136].

The adhesion molecule fibronectin occurs in blood plasma in soluble form, but also as an insoluble ECM molecule, which is one of the important structural components of the BM [137]. FN is a homodimer consisting of two chains linked by a disulfide covalent bond. FN chains are divided into repeating protein domains of type I, type II, or type III. The domains interact with integrin receptors, contain binding sites for integrins, heparin, chondroitin sulfate (CS) proteoglycans, and COLs, and can crosslink to form larger ECM complexes [138].

The integrin family includes the FN receptor, which is characterized by recognizing the RGD sequence in its ligands. Analysis of cell spatial organization and cell adhesion revealed a crucial role for adhesive ligands such as the Arg-Gly-Asp (RGD) peptide. The RGD peptide is a well-characterized adhesive domain of ECM components, mediating the binding of FN, which facilitates crosslinking of the COL matrix, and interacting with the alpha 5 beta1 integrin receptor via the RGD motif present on COL [139,140]. The dimeric FN molecule plays a significant role in the BM environment in regulating the erythroid lineage of cells. This is particularly evident in the stage that follows the early stage, which is influenced by erythropoietin. Erythroid progenitors have a severe affinity and binding to FN, which changes during maturation, when adhesion to FN is significantly reduced [141]. An active role for FN in the BM has been detected in the differentiation of megakaryocytes and mature platelets. BM MSCs produce FN in significant amounts. FN has been detected in vitro under simulated BM conditions when culturing BM MSC environments [142].

The tenascin family includes four members: tenascin-C, -R, -W, and -X. Tenascin-C, which is typical of the hematopoietic environment, is expressed in the BM in a physiological state. Tenascin-W is expressed in the BM during metastatic processes. The tenascin-C molecule of six subunits is assembled at its N-terminal ends in a structure known as a “hexabrachion” [143]. Tenascin-C is thought to play a significant role in hematopoietic cell development, particularly in erythropoiesis [144]. Osteopontin is a protein involved in the formation of the BM matrix, which belongs to the “small integrin-binding ligand N-linked glycoprotein”. It is expressed by osteoblasts at the border of bone tissue. It forms an adhesive substrate for HSCs after hematopoietic graft transplantation by thrombin, which cleaves the osteopontin fragment. This makes the binding site for integrin alpha9beta1 accessible on HSCs [145]. Osteonectin, or Secreted Protein Acidic and Rich in Cysteine SPARC, is a matrix glycoprotein of acidic nature. Osteoblasts and BM ECs have significant expression of osteonectin. It has an impact on the regulation of HSC proliferation [146]. The periostin molecule has been confirmed in osteoblast culture. In BM, periostin has been detected in osteoblasts and MSCs. Its role in B-lymphopoiesis has been reported, and a decrease in periostin affects the development of B-cells [147].

### 3.2. Collagens (COLs)

The collagen family is one of the most abundant proteins. COL is generally composed of three polypeptide alpha chains, either homo- or heterotrimeric, depending on the class. The triple structure is dextrorotatory, in the form of a helix, and at every third position the amino acid glycine is present, followed by proline or hydroxyproline [148]. The main role of COLs is their structural and mechanical function. COLs are involved in the regulation of cell adhesion, proliferation, migration, and differentiation [149]. The expression of several types of COL1, COL3, COL4, COL5, COL6, and COL14 has been demonstrated in BM at the protein level [150,151]. A significant amount of COL1 is contained in the bone matrix of BM. In BM, osteoblasts and BM stromal cells express type COL1. Its synthesis is influenced by TGF-β1 in MSC [152]. Several studies have noted that COL1 occurs in a filament structure in the BM [153,154]. COL1 is considered a suitable substrate for the adhesion of HPCs. In vitro, B-lymphoid and plasma cells bind to COL1 via the syndecan receptor. In vitro culture of HSCs coated with COL1 reduces proliferation and induces differentiation. This effect suggests an effect of COL1 on maintaining the quiescent state of stem cells in the BM niche [155]. A different study reported a stimulatory role of type COL1 in vitro on MSC proliferation with a significant effect on osteogenic differentiation [156]. In BM, COL3 is localized as sporadic fibrils and near arterioles or periosteal areas. COL3 is involved in trabecular bone development and osteoblastogenesis, and is a non-adhesive substrate for hematopoietic cell types [157]. In the BM environment, COL4 is localized in the endosteal, periarteriolar region, and in the sinusoids of the BM [153]. COL4 has a stimulatory effect on prothrombocytes in sinusoids during hematopoiesis, while megakaryocytes express COL4 [158]. COL6 forms microfibrillar structures in extrasinusoidal spaces. BM also contains COL4, COL6, COL9, COL10, COL14, and COL18 with diverse functions [2].

### 3.3. Proteoglycans

Proteoglycans are another structural component of the ECM niche of the BM. Proteoglycan molecules are composed of core proteins and glycosaminoglycan (GAG) side chains. The side chains are formed by repeating disaccharide units, with the heparan sulfate proteoglycan (HSPG) family in particular having a significant function in the HSC niche. Perlecan belongs to the HSPG family and contains a core protein with three GAG side chains. Perlecan, also known as HSPG-2, is synthesized by MSCs in the BM niche and acts on the internal connective tissue architecture of the surrounding environment. HSPG2 is highly expressed in the BM and in vitro in an environment mimicking hematopoiesis. It has an antiadhesive effect on HSCs, but an adhesive effect on endothelial and fibroblast cells [159]. Notably, HSPG2 may play a role in regulating hematopoiesis through its ability to bind granulocyte–macrophage colony-stimulating factor (GM-CSF) [150]. Another major GAG component in the BM is hyaluronic acid (HA), which contains unsulfated linear GAG. HA is composed of disaccharide units of glucuronic acid and N-acetylglucosamine and has binding activity for several receptors and growth factors of the HSC niche. The cell-surface glycoprotein CD44 functions as a receptor for HA. BM MSCs and Lin-Sca+Kit+ HSC populations participate in HA synthesis under the influence of IL-beta [160]. The receptor for hyaluronan-mediated motility (RHAMM) affects haematopoietic stem and progenitor cells (HSPC) motility, which is essential for the function of HA in HSPC mobilization and proliferation. CD44 acts on HSPC adhesion [161] (Figure 3).

## 4. Modeling the Hematopoietic Niche

A key challenge in bone marrow research is the development of an ex vivo model that accurately replicates theBM environment. Establishing a representative simulation of the medullary microenvironment of hematopoiesis (MMH) is crucial for investigating both physiological and pathological aspects of hematopoiesis [1,162]. To maintain HSC stemness in vitro, it is necessary to create a niche environment that is as similar as possible to the BM. The multifactorial environment can be mimicked by multiple cell populations, most commonly MSCs, osteoblasts, which cooperate with HSCs and the presence of growth factors and cytokines [163]. Loss of multipotency of HSCs has been reported after their in vitro culture, outside the natural environment of BMM. The presumed reason is the absence of a hematopoietic stimulatory niche [164]. The main goal is to modulate not only two-dimensional (2D) but also three-dimensional (3D) cell cultures using biomaterials that mimic the BM niche in the form of hydrogels, scaffolds, in combination with ECM components. MSCs play a significant role in the function and stemness of HSC/HPC [9]. With their ability to differentiate into osteoblasts, adipocytes, and CAR cells, they are an essential component of BMM [4]. MSCs, as precursor cells of mesenchymal cell types in the hematopoietic niche, such as adipocytes, osteoblasts, and fibroblasts, represent important and defining cell types of this niche and are suitable as a cell population for modeling the BM niche in vitro [165].

### 4.1. Two-Dimensional Suspension Cultures of Human Hematopoietic Stem and Progenitor Cells (HSCs/HPCs) and Co-Culture with Mesenchymal Stem Cells

Long-term culture and maintenance of HSC/HPC is challenging, and a suitable in vitro model of the BM niche is still being sought [163]. In vitro culture of HSC/HPC is still an open question and requires preservation of stemness properties. For the culture and maintenance of HSC/HPC, a specific environment of cytokines, growth factors, and non-hematopoietic cell populations of heterogeneous stromal cells is essential [163]. As a medium for the culture of HSC from umbilical cord blood (UCB), Iscove’s modified Dulbecco’s medium (IMDM), enriched with fetal bovine serum and combinations of cytokines SCF, Granulocyte–macrophage colony-stimulating factor (GM-CSF), IL-3, TPO, and IL-6 were mainly used. The expansion of CD34+ cells in a static culture lasting 7 days appeared to be the most suitable, thus demonstrating the feasibility of short-term liquid culture with cytokine stimulation [166].

A simpler simulation of the BM niche involves suspension culture with the addition of BM cytokines, most commonly SCF, thrombopoietin, FMS-like tyrosine kinase 3 ligand (Flt3L), angiopoietin-like proteins, and IL-6 [164,167]. Recently applied cytokines such as nerve growth factor (NGF) and IL-11 help maintain HSC pluripotency [168]. Other molecules with effects on maintaining stemness and inhibiting HSC differentiation have also been tested, such as nicotinamide [169] and prostaglandin E2 (PGE2) [170].

McNiece et al. developed a multi-phase culture method with initial expansion of UCB CD34+, CD133+, and subsequent co-culture system with BM MSCs with UCB hematopoietic progenitors [171]. After isolation, CD133+ and CD34+ were cultured in a Miltenyi CliniMACS liquid culture system and culture medium with a cocktail of cytokines SCF, IL-3, IL-6, and G-CSF. MSCs, a population of non-hematopoietic cells of KD and other tissues, were cultured as adherent cells under standard culture conditions with the addition of fetal bovine serum in Alpha modification Minimum Essential Medium (alpha MEM) [172]. MSC KD was passaged after a 70–80% monolayer and then cocultured in culture flasks for an additional 2 weeks, after which ex vivo expanded UCB CD34+ and CD133+ were added to the system. The coculture system was designed based on the ability of MSCs to support the proliferation of hematological progenitors by producing growth factors and adhesion molecules [173]. Overall, they achieved a 6-fold increase in CD34+ cells after coculture [171]. The ex vivo expansion strategy is also applicable to BM hematopoietic progenitors. The main benefit of these culture procedures is to enrich the knowledge about the BM niche and cooperating cell population and, last but not least, to increase the quality of transplantable grafts in bone marrow transplantation and also HSC/HPC after peripheral blood mobilization [174]. DeLima et al. tested the possibility of increasing the quality of transplantable cord blood grafts through coculture with MSCs. They used cord blood, which elicits a lower immune response, as a source for transplantation in recipients with hematologic cancers. The MSC donors were haploidentical family members. MSCs were isolated from the BM of donors and cultured in vitro in alpha MEM supplemented with bovine fetal serum. After the formation of a monolayer of MSCs, cord blood was added. One week after coculture, non-adherent cells were harvested and administered intravenously to patients after washing. The authors compared graft engraftment with a group that received cord blood without expansion. The authors noted a statistically significant increase in CD34+ progenitors and an increased proportion of monocytes and granulocytes that were expanded in coculture with MSCs. The main purpose of the work was to improve hematopoiesis in the monitored patients [175].

#### Surfaces with ECM Component Properties

Considering the cell-to-cell contact of HSCs through cell surface molecules, the use of coating culture surfaces with ECM molecules such as laminin (LM) or FN in vitro is being tested. Enhancing the properties of culture surfaces by leveraging interactions between ECM molecules and cells serves to mimic the BM niche. It is hypothesized that the specific combination of applied molecules may influence HSC differentiation. LM-coated surfaces were found to be stimulatory for the differentiation of megakaryocyte progenitors, and FN-coated surfaces appeared to be stimulatory for erythroid progenitors. The authors tested a multicomponent coating composed of FN, LM, COL1, and COL4, which acted to stimulate myeloid differentiation in a population of cord blood HPCs compared to a coating without COL [176].

### 4.2. Three-Dimensional Culture

The BMM has the role of maintaining homeostasis by allowing communication through growth factors in the 3D space created by the ECM components. A three-dimensional culture system with a spatial cell arrangement that mimics the in vivo tissue environment is more suitable for the complex multicellular environment of the BM. BM stromal cell spheroid culture mediates cell–cell interactions and promotes ECM production. Spheroid generation is available by several methods such as “hanging drop” techniques [177] and magnetic levitation [178]. A spheroid culture system was described in which MSCs were labeled with magnetic particles and subsequently assembled via magnetic levitation. The resulting spheroids exhibited characteristic BM MSC phenotypes, including expression of nestin and Stro-1. MSC spheroids, in combination with osteoblast populations and ECs, mimicked characteristic regions of the BM, namely, the endosteal and perivascular niches [178]. The endosteal niche, which is formed by the interface between bone tissue and BM, is formed by a layer of osteoblasts and a small population of osteoclasts [179]. Spheroids appear to be a promising way to simulate the BM niche and co-culture MSCs and HSCs [1]. A monolayer of MSCs enhances HSC expansion in vitro, but the maintenance of HSC stemness is a problem [175]. Several studies have shown that HSCs/HPCs expanded on an MSC monolayer lose the ability to colonize long term in patients [180,181]. Futrega et al. tested a 3D spheroid culture, which more naturally recapitulates BM [182]. MSCs were isolated from the BM of healthy donors and monolayer cultured in DMEM medium with fetal bovine serum, and their mesenchymal phenotype was confirmed by flow cytometry. In their experiment, they compared a 2D co-culture of MSCs with CD34+ progenitors and a spheroid culture of both populations. CD34+ progenitors were isolated from umbilical cord blood (CB) using a selective isolation method using magnetic particles bound to the CD34 MicroBead antibody. Spheroids consisting of MSCs and CD34+ HUVEC progenitors were prepared on microwell plates. Increased numbers and the expansion of CD34+ progenitors after one week were noted compared to culture without the MSC population. Overall, an increase in the number of CD34+CD38− cord blood cells was observed in spheroid culture [182].

#### 4.2.1. Biocompatible Synthetic and Natural Scaffolds

Biocompatible scaffolds are fully adapted for a 3D cell culture. The porous scaffold system creates a suitable structure for in vitro BM simulation, such as poly (D, L-lactide-co-glycolide) (PLGA) [183], polyethersulfone (PES), and non-woven polyethylene terephthalate (PET) fabric [184]. Synthetic scaffolds are suitable for their surface, which is adapted for cell adhesion, porosity, and permeation of culture medium containing trophic factors. Scaffolds that do not contain specific cell-binding sites can be coated with ECM molecules such as COL, FN, and laminin to increase their biocompatibility [185]. Another option for in vitro BM simulation and cell carrier applications is decellularized ECM matrices. The said scaffolds create a complex environment for HSCs and MSCs, whereby the production of CXCL12 and SCF was confirmed, and thus the microenvironment is similar to BM [186].

#### 4.2.2. Hydrogels

Hydrogels are prepared with an emphasis on biomimetic design, especially in terms of porosity, biochemical composition, and allowing for intercellular contact with the activity of growth factors. Stem cells cultured in hydrogel medium retain their stemness, but proliferation is reduced [187]. Encapsulating cells in gels creates a microenvironment that approximates the simulation of BM. Natural hydrogels, hyaluronic acid (HA) [188], COL [189], alginate [190], and fibrin [183] are the most commonly used. Co-cultures of MSCs as cells in hydrogels in combination with HSCs simulating the BM niche have also been described [191]. Animal-derived “Matrigel” provides a 3D environment for stem cell culture under highly stimulating conditions. It is a product of the isolation of ECM proteins originally from the mouse Engelbreth–Holm–Swarm (EHS) tumor with a complex composition of laminin, COL4, proteoglycan, heparin sulfate, and growth factors TGF beta, FGF, EGF, and PDGF [192]. In Matrigel, specific epithelial cells were polarized [193]. The authors Baghaban followed the culture of MSCs isolated from BM in Matrigel, monitoring several parameters such as proliferation and differentiation into osteoblasts and alkaline phosphatase expression. They confirmed the ability to stimulate osteogenic differentiation of MSCs in a Matrigel environment. Differentiated osteogenic cells have the potential to support HSCs and hematopoiesis [193].

#### 4.2.3. Synthetic Gels with Scaffolds

Despite their advantages in supporting cell culture, natural gels exhibit less favorable mechanical properties and limited standardization, which can hinder their reproducibility and suitability for broader applications. Synthetic gels have the advantage of reproducible possibilities and the creation of the desired mechanical properties. The materials traditionally used are poly(l-lactic acid) (PLLA), poly(ethylene oxide) (PEO), and poly(ethylene glycol) (PEG) [194,195,196]. The combination of synthetic gel and growth factors has been reported to preserve the pluripotent state of HSCs for a longer period of time. Enrichment of gel methacrylate (GelMA) hydrogels with SCF has been shown to maintain HSCs’ stemness during a week of culture [189]. Synthetic polymers can be suitably modified with RGD peptides, resulting in improved cell adhesion properties [197]. Trujillo and colleagues developed a hydrogel based on FN with defined stiffness and suitable biodegradability and enriched with VEGF and BMP2 [198]. Engineered biomaterials must be capable of appropriately releasing growth factors in a controlled manner to exert localized biological effects. In the 3D environment, the ability of fibrillogenesis of molecules synthesized by active cells is confirmed. The polymer material poly(ethyl acrylate) (PEA) acts on FN crosslinking and the possibility of growth factors’ diffusion and activity [199].

Macroporous hydrogels represent an innovative strategy for mimicking the BM microenvironment and promoting the formation of MSC spheroids. The complex morphology of the BM consists of an interconnected network of sinusoids and ECM, within which heterogeneous cell populations dynamically interact. The size of the “pores” in the BM is in the range from several microns to several millimeters and is determined by the structure of glycoproteins and proteoglycans, while the structure is adapted for the movement of cells and the transfer of growth factors. Hydrogel design for in vitro BM formation favors its porous structure and is suitable for MSCs culture. Macroporous hydrogels can be prepared by template and phase separation of liquids with PEG and dextran, PEG acting as the percolating phase and dextran forming non-percolating droplets. The dextran phase acts as a “soft template” for the formation of macropores [200]. The 3D BM niche complex forms an interface between cell populations, growth factors, cytokines, and biomaterials in vitro. The 3D model simulates the bone marrow environment with a combination of a rigid component representing bone structure and a softer marrow region populated by cell populations [201]. Ravichandran tested a bone marrow adipose tissue (BMAT) model based on a combination of a GelMA scaffold with a hydrogel/medical-grade polycaprolactone (mPCL) scaffold composite seeded with human BM stromal cells. The soft GelMA material was seeded with MSCs in combination with a tubular, macroporous mPCL scaffold that is designed for mechanical loading. The bioreactor system in which the samples were placed for three weeks in adipogenic differentiation medium provided a suitable environment during which cell viability was maintained and the cells formed lipid droplets in the cytoplasm [202].

#### 4.2.4. Bone Marrow-on-a-Chip

The most comprehensive BM model is the chip-on system. It allows for the co-cultivation of the relevant cell environment and the simulation of a BM niche containing a population of MSCs and ECs that maintain mutual contact and produce ECM in a microfluidic system. Sieber et al. present a BM niche model that allows several weeks of HSC lineage maintenance. The complex combines a scaffold with a microfluidic system and a cellular component of MSCs BM, which was successfully sustained for more than a month in these dynamic setups. The result of the model is the possibility of studying intercellular interactions and, from a practical point of view, also testing and screening drugs [203]. Bruce et al. created a BM culture system environment consisting of lymphoblastic cells, MSCs, and osteoblasts in a 3D COL1 matrix connected to a perfusion system, thereby creating variable combinations of cell–matrix interactions [204]. The results of the above studies assist in understanding the causes of hemato-oncological diseases and may also contribute to the testing of treatment methods for these diseases (Figure 4). Kefallinou et al. describe the construction of an in vitro BM model, namely, Bone-marrow-on-a-chip (BMoC). The proposed BMoC is a scaffold-free device that combines microfluidic technology and co-culture of fluorescently labeled human HSPCs and MSCs. The BMoC perfusion model was constructed as a PDMS or poly(ethylene terephthalate) (PET) membrane in the form of a chip with the possibility of microscopic observation of cells in MSC-HSPC co-culture, which is placed in a culture dish with growth medium. Adhered MSCs on the PET membrane maintained their spindle-shaped morphology and were evenly distributed. In the case of PDMS membranes, MSCs did not have a homogeneous distribution and accumulated regionally. The observed HSPCs were homogeneously distributed. The biomimetic BM system was described as an in vitro simulation of a hematopoietic niche near the perivascular niche [205].

#### 4.2.5. Biomimetic 3D Model of Bone Marrow

Bosh-Fortea et al. addressed the topic of ex vivo expansion of HSC/HPC in a biomimetic artificial BM culture system, which aimed to most closely approximate the need for hematopoietic cells in the organism. The presented BM model is based on a bioemulsion in which protein nanosheets are stabilized, which influence MSC adhesion and phenotype maintenance during long-term cultivation. Bioemulsions were stabilized by homogeneous poly(L-lysine) (PLL) nanosheets, in combination with FN. MSCs were added first, and after their one-week expansion, HSCs/HPCs were added and cultured in the system for another two weeks. They reported on the design of artificial BM niches, which recapitulate the microstructure, mechanical conditions, and cell populations of BM niches in vivo. Bioemulsions are suitable for MSC expansion and are adapted for hematopoietic stem cells. Microscopic analysis of microdroplets revealed that MSCs with a characteristic phenotype adhered to the droplet surface and proliferated. The MSC microstructure contained F-actin cytoskeleton at the interface with nanosheet and liquid interfaces and beta1-integrin as formation focal adhesions. The authors characterized the expression of cytokine adhesion molecules in the bioemulsion after more than two weeks of 3D MSC culture and detected the expression of SCF, IL-6, angiopoietin, thrombopoietin, VCAM, N-cadherin, and Jagged 1. Furthermore, Nestin expression was confirmed in MSCs over two weeks of culture, and a subset of LeR+ MSC cells supporting HSC survival and stemness was detected. Interestingly, the growth of the culture is a slight increase in the percentage of CD34+ CD38− HSC/HPC. Microscopically, after 15 days of co-culture, HSCs/HPCs were identified in the central part of the microdroplet. Immunohistochemical staining demonstrated the interaction of HSCs/HPCs and MSCs, specifically the binding of SCF to the c-kit receptor of HSCs and MSCs produced CXCL12 with binding to CXCR4 on the surface of HSCs/HPCs [206].

Bourgine et al. presented a 3D biomimetic model of BM in a bioreactor system in a perfusion system using hydroxyapatite scaffolds, human MSCs, and human umbilical cord progenitors (HUVEC) CD34+, structurally similar to bone tissue. Human MSCs were initially cultured in a perfusion system in proliferation medium on the scaffold, where they were differentiated into osteoblastic lineages. The perfusion system provided continuous exchange of the culture medium with a continuous supply of nutrients to the cells and removal of metabolic waste products. HUVEC CD34+ were added to the stabilized 3D system consisting of a scaffold with seeded and proliferating MSCs. Scanning electron microscopy results confirmed a homogeneous layering of ECM containing cells of fibroblastic morphology, presumably MSCs, and round-shaped HUVEC cells. Finally, after a month of culture, a gel-like tissue formed de novo was observed. The composition of the generated ECM after one month of culture was tested, and the presence of COL1, COL4, fibronectin, and osteocalcin, which are characteristic components of BMM, was confirmed by immunohistochemistry. Advanced biomimetic BM models take into account the complexity and intricacy of the cooperating populations of hematopoietic and stromal stem cells, as well as the biochemical and physical properties of BMM [142] (Table 3).

## 5. Conclusions

The BMM and the properties of HSCs are currently being studied in detail. The concept of the bone marrow niche in vitro is, in most cases, a combination of adherent MSCs with a characteristic phenotype and co-cultured CD34+ HSCs/HPCs [206]. The important role of MSCs and differentiated stromal populations for the function of hematopoietic stem cells is to create an environment that supports stemness and ensures the expansion of HSCs/HPCs in vitro by interactions between cells. Biological processes in the bone marrow are mediated by the influence of several cytokines, among which CXCL12 and SCF are important with their influence on HSCs. These are expressed to a greater extent by subpopulations of MSCs and CAR cells [207]. The advantage of advanced BMM models is that the creation of an environment containing populations of HSC/HPC and MSC stromal cells with variable combinations of 3D spheroids and solid scaffolds in a simulated environment with continuous culture medium exchange and cytokine exposure allows the investigation of both physiological and pathological processes in the bone marrow.

## Figures and Tables

**Figure 1 ijms-26-09036-f001:**
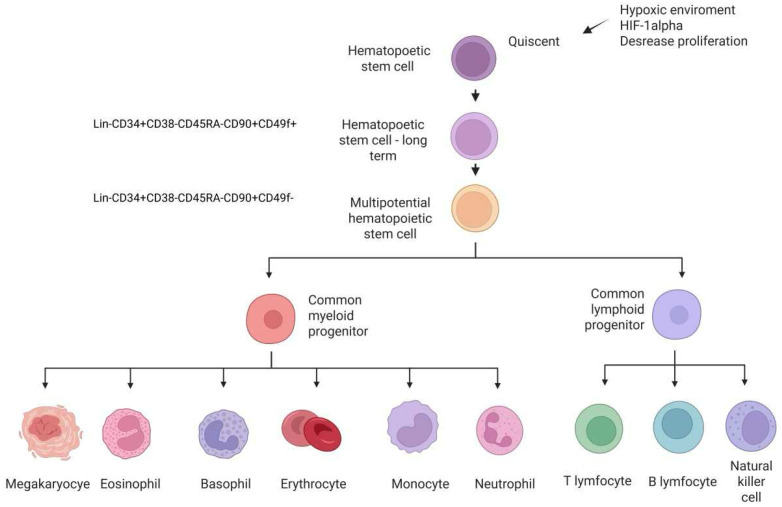
Differentiation of bone marrow (BM) hematopoietic cells. BM is a rich source of hematopoietic stem cells (HSCs), multipotent progenitors that are responsible for the generation and maintenance of cellular elements of the blood. During quiescence, the BM acts as a reservoir for quiescent HSCs. HPCs ensure the maintenance of the number and steady state of the blood count and the activation of stem cell clones, which differentiate into two main types of progenitors, myeloid and lymphoid. Myeloid progenitors give rise to cells of the myeloid lineage during hematopoiesis, which include granulocytes: eosinophil, basophil, neutrophil, and monocytes: macrophages, dendritic cells, megakaryocytes, and erythrocytes. Lymphoid progenitors tend to differentiate into T-lymphocytes, B-lymphocytes, and NK cells. Hematopoietic differentiation is controlled by extrinsic cytokines and intrinsic transcription factors. Created in https://BioRender.com.

**Figure 2 ijms-26-09036-f002:**
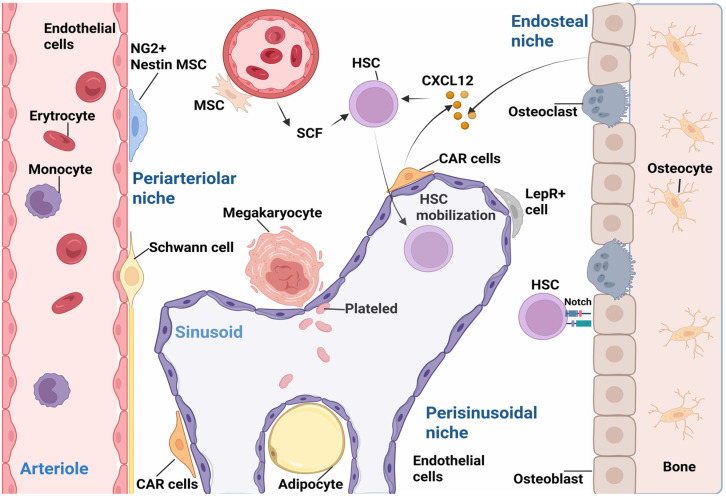
The bone marrow microenvironment (BMM), which contains a perisinusoidal niche with numerous populations of heterogeneous cells, including mesenchymal stem cells (MSCs), perivascular stromal cells, endothelial cells (ECs), macrophages, CAR cells, Nestin+ and neuron-glial antigen 2 (NG2+) cells, and Schwann cells interacting with hematopoietic stem cells (HSCs). The main factors influencing HSCs include stem cell factor (SCF) and stromal cell-derived factor 1 (SDF-1), also known as CXC motif chemokine ligand 12 (CXCL12). The endosteal niche is composed predominantly of a population of osteoblasts and a smaller number of osteoclasts. Resting HSCs are localized in the vicinity of the endosteal niche. Created in https://BioRender.com.

**Figure 3 ijms-26-09036-f003:**
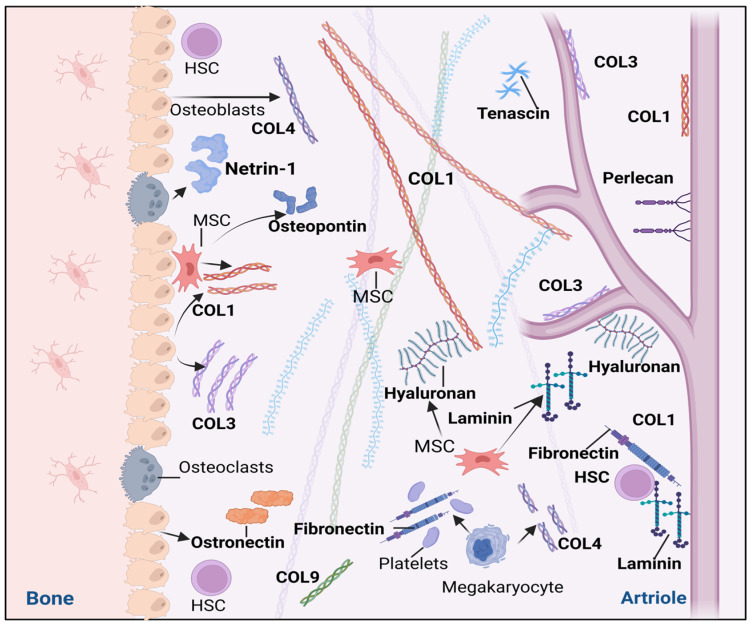
The microenvironment in the bone marrow (BM) is formed by the extracellular matrix (ECM), a network-like environment that contains hematopoietic and non-hematopoietic cells. The BM matrix is composed of structural macromolecules such as collagens (COLs), fibronectin (FN), laminin, and proteoglycans. Hematopoietic stem cells (HSCs) interact with the surrounding network matrix through receptors, namely, cell adhesion molecules (CAMs). A significant component is glycoproteins, namely, netrins, which cooperate with laminin molecules through adhesion. FN acts adhesively as a structural component of the bone membrane. The tenascin-C molecule occurs in a “hexabrachion” structure. COLs have mechanical and structural functions. BM contains COL1, COL3, COL4, COL5, COL6, and COL11. Hyaluronic acid (HA) contains GAGs with the ability to bind to the surface glycoprotein CD44 on mesenchymal stem cells (MSCs). Glycosaminoglycans provide binding sites for growth factors that influence the metabolism of HSCs and related BM cell populations. Created in https://BioRender.com.

**Figure 4 ijms-26-09036-f004:**
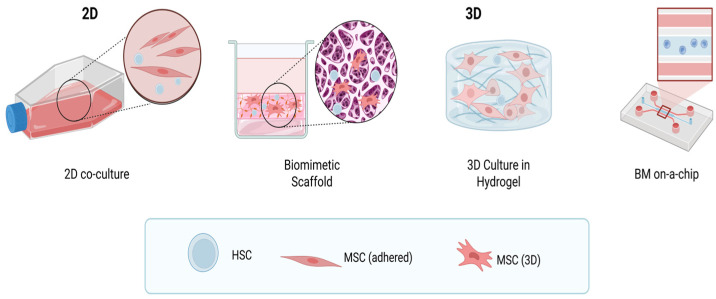
In vitro modeling of a hematopoietic niche. A simpler simulation of the bone marrow (BM) niche involves suspension cell culture. Biocompatible scaffolds are fully adapted for a 3D cell culture. The porous scaffold system creates a suitable structure for in vitro BM simulation. The macroporous hydrogel enables simulation of the BM environment and the formation of mesenchymal stem cells (MSC) spheroids. The chip-on system allows for the co-cultivation of the relevant cell environment and the simulation of a BM niche containing a population of MSCs, endothelial cells (ECs) that maintain mutual contact and produce extracellular matrix (ECM) in a microfluidic system. Created in https://BioRender.com.

**Table 1 ijms-26-09036-t001:** Phenotypes of surface marker human hematopoietic stem and progenitor cells (HSCs/HPCs) in bone marrow (BM), peripheral blood, and cord blood. Abbreviations. Long-term culture-initiating cells (LTC-ICs), Colony-forming cells (CFCs), and hematopoietic progenitor cells (HPCs).

Panel of Surface Marker Human Hematopoietic and Progenitor Stem Cells	Cell-Type Specificity	Localization	References
CD34+ cells	Heterogeneous stem cells include committed progenitors	BM and peripheral blood	[31,32]
CD34+ CD38−	LTC-IC, CFCs	BM	[39]
Lin− CD34+ CD38+	In HSCs, expression of CD38 is correlated with increased differentiation	BM and peripheral blood	[42]
Lin− CD34+ CD38− CD45RA− CD90− CD49f−	HPC	BM, peripheral blood, cord blood	[40]
Lin− CD90+ CD45RA− CD71−	HSC/HPC	peripheral blood mobilisation	[43,44]
Lin− CD34+ CD38− CD45RA− CD90+ CD49f+	Long-term repopulating hematopoietic stem cells	BM and cord blood	[22]

**Table 3 ijms-26-09036-t003:** Types of bone marrow in vitro models with advantages and disadvantages of different cell culture procedures. Abbreviations: bone marrow (BM), Dulbecco’s Modified Eagle Medium (DMEM), extracellular matrix (ECM), fetal calf serum (FCS), Fms-like tyrosine kinase 3 ligand, (Flt3-Ligand), hematopoetic stem cells/progenitor cells (HSC/HPC), interleukin (IL), Iscove’s modified Dulbecco’s medium (IMDM), mesenchymal stem cells (MSC), poly(ethylene terephthalate) (PET), polycaprolactone (PLC), poly(lactic-co-glycolic acid) (PLGA), stem cell factor (SCF), thrombopoietin (TPO), umbilical cord blood (UCB).

Type Culture/Cell Types	Conditions Cell Cultures	Advantages/Disadvantages	References
Suspension culture Human umbilical cord blood (UCB) CD34+	UCB CD34+ cells: Iscove’s modified Dulbecco’s medium (IMDM), fetal calf serum (FSC) stem cell factor (SCF) thrombopoietin, FMS-like tyrosine kinase 3 ligand (Flt3L) angiopoietin-like proteins (ANGPTLs), IL-6	Advantages: expansion of UCB CD34+ cells easy-to-use protocol high reproducibility Disadvantages: loss stemness	[164,167]
Suspension hematopoietic cells and monolayer mesenchymal stem cells (MSC) co-culture Human UCB HSC/HPC CD34+ and MSC	UCB CD34+ cells: IMDM, FCS, SCF, Granulocyte-macrophage colony-stimulating factor (GM-CSF), IL-3, Thrombopoietin (TPO), IL-6 MSC: alpha MEM, FCS	Advantages: co-culture ability isolation and maintaining viability during cultivation large number of cells in subculture increase in CD34+ Disadvantages: loss stemness	[163,166,171,180]
Surfaces for culture with coating with COL1, fibronectin, laminin human UCB HSC/HPC CD34+	UCB CD34+ cells: IMDM serum-free medium substitute albumin/insulin/transferrin	Advantages: ex vivo expansion of UCB CD34+ cells unlimited cell growth large numbers of cells by subculturing Disadvantages: absence of BM stromal cells	[176]
3D culture spheroid techniques human UCB CD34+ HSC/HPC bone marrow (BM) MSC	UCB CD34+ cells: Serum-free medium for hematopoietic cells (SFEM)	Advantages: BM stromal cell spheroids improve cell–cell interactions and promote ECM production mimicked the endosteal and perivascular niches Disadvantages: demanding cultivation system	[177,178,179]
	BM MSC: DMEM low glucose, human thrombocyte lysate, L-glutamine, HEPES sodium salt		
Scaffolds PCL, PLGA, fibrin a collagen UCB HSC/HPC CD34+ UCB MSC	UCB CD34+ cells: SCF, thrombopoetin, fibroblast growth factor-1, angiopoietin like-5, insulin-like growth factor binding-protein 2 UCB MSC: alpha MEM medium, FCS, insulin–transferrin–selenic acid, linoleic acid	Advantages: culture CD34^+^ cells expanded on 3D fibrin scaffolds with UC MSC 3D scaffold PLGA meshes 3D fibrin scaffolds with stromal support for expansion of CB CD34^+^ cells in the presence of cytokine supplementation with UCB MSC HSC adhesion to fibrin scaffold, as in the HSC niche Disadvantages: PLGA meshes did not support HSC expansion	[183]
Bone marrow-on-a-chip PET membrane in the form of a chip with cells in UB MSC and HSC/HPC CD34+ co-culture	UCB CD34+ cells: serum-free medium for expansion of hematopoietic cells with SCF, Flt3-Ligand UB MSC: alpha MEM, FCS	Advantages: microfluidic system with passive perfusion with contact HSC/HPC and MSC culture cells and produce ECM, increase the population of HSCs/HPCs Disadvantages: technically demanding culture system	[203,205]
Biomimetic 3D model of bone marrow co-culture of UCB CD34+ and UCB MSC	UCB CD34+ HSPC and UCB MSC: culture growth medium, SCF	Advantages: Biomimic perfusion system with architecture like BM De novo ECM formation with COL 1, COL 4, fibronectin, osteocalcin Suitable for studying the bone marrow niche Disadvantages: technically demanding culture system	[206,142]

## Data Availability

Data is contained within the article. The original contributions presented in the study are included in the article; further inquiries can be directed to the corresponding author.

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
