# Peer review of "Interactions of Hematopoietic and Associated Mesenchymal Stem Cell Populations in the Bone Marrow Microenvironment, In Vivo and In Vitro Model"

_ijms, 2025, doi:10.3390/ijms26189036_

Round 1
Reviewer 1 Report
Comments and Suggestions for Authors
This paper is very difficult to follow and might benefit from major revision by a native speaker. Above all, the manuscript lacks novelty. Without any doubt, interactions between HSCs and the microenvironment play a vital role in regulating murine and human hematopoiesis in health, aging, and in disease. Within this context, development of an appropriate 3-D bone marrow niche model, and a model that could reversibly alter its stiffness, is essential. The present manuscript superficially touches on a plethora of topics around these very complex and convoluted processes, but provides little incremental information or direction for the readers.
Comments on the Quality of English LanguageThe manuscript is difficult to follow. In the Abstract alone, there are numerous expressions that need to be corrected for the manuscript to be readable or comprehensible:
- Line 14 "Hematopoietic stem cells (HSCs) take place in niches......
- Line 19 "The heterogeneity of the niche environment is conditioned ....
- Line 26 "Development of a suitable simulation of the BMM is essential....
and these continue throughout the manuscript.
Author Response
Dear reviewer, the corrections of our manuscript are included in the attachment.
Thank you for your time,
best regards,
Bacenkova

Reviewer 2 Report
Comments and Suggestions for Authors
The text is poorly structured and is a collection of data that is not united by a common idea. The review uses many outdated sources that do not take into account modern data. Of the 205 sources, one article was published in 2025, 5 in 2024, 4 in 2023, 5 in 2022, and 4 in 2021. For a review of the hematopoietic system and bone stromal microenvironment, topics on which new data are continually emerging, less than 10% of sources are younger than 5 years old - this is too few. Bone marrow stromal cells vary greatly between in vitro and in vivo data, between humans and mice, so it is important to note the object of study when describing their characteristics. In this article, they are mixed together, which creates a false picture. The history of the research is described in excessive detail, as the latest results with opponents and supporters. The narrative is structured in a chaotic manner, the authors jump from stromal to hematopoietic cells and back without transitions, which makes it impossible to follow the logic of the story. Some detailed remarks for example:
Line 14: The most undifferentiated HSCs do not express СВ34+ (https://www.sciencedirect.com/science/article/pii/S0301472X20301818), so the wording of the first sentence should be revised. The expression most often used in literature is CD34+ hematopoietic stem/progenitor cell (HSC/HPC).
Line 18: Revision is needed: bone marrow niche include not only hematopoietic cells and MSCs, but also many differentiated stromal cells, nerve and endothelial cells.
Line 38: Please, change BM-HSCs to HSCs.
Line 38: The reduction BM was introduced twice.
Lines 40-48: It sounds like the results are from the same authors, but they are not.
Line 45: In this work “STRO-1 did bind to subpopulations of cells expressing CD34, CD44, and CD29”
Line 48: The reference 6 is not correct.
Line 49: The abbreviation “ECM” was introduced for the first time and was not deciphered.
Line 48: “In the BM niche model…” Which model is meant? The cultural one, presumably described in the previous sentence, or the theoretical notion of a niche?
Reference 10: The article name is “A direct measurement of the radiation sensitivity of normal mouse bone marrow cells. 1961”
Lines 52-60: Till and McCulloch's work demonstrated the existence of a hematopoietic stem cell, not just a hematopoietic precursor. They used chromosomal aberrations as a cytogenetic marker, which allowed them to prove that all the cells in a colony were descendants of a single cell.
Lines 66-70: “The HSC niche is generally described as perisinusoidal, near the sinusoids, and the endosteum region mainly provides a niche for HSCs [15,16]. This confirmed the assumption of the occurrence of an osteoblastic niche in the endosteum.” The second sentence does not follow from the first. The generally accepted description cannot prove anything.
Line 73: In no niche theory is the stromal component limited to one cell type.
Line 77: The previous text does not make it clear why MSC?
Line 83: “Approximately 1.3×1012 BM cells are released into the bloodstream daily. Of these, about
200 are erythrocytes and 70 are neutrophils” And what are other 1.2×1012 cells?
Line 85: BM consists of two distinct forms: hematopoietic tissue and supportive marrow stroma. Both cell types are formed in solid cords that are separated by sinusoids. – Hematopoietic and stromal cells are not separated in the bone marrow
Line 99: “Human HSCs originate from the endothelium that lines blood vessels in both fetal and non-fetal tissues” - Who is this shown by? Reference?
Lines 98-111: This section needs to be revised, there are many good reviews based on modern concepts, for example https://doi.org/10.3390/ijms22179231
Line 115: Write “the latest data” citing a 2018 article is not correct. Seven years have passed.
Author Response

(The authors gave the same response as above.)

Round 2
Reviewer 1 Report
Comments and Suggestions for Authors
The main challenge for the manuscript is that it lacks novelty. Despite all the various changes that the authors have made, it presents an extensive narrative of knowledge that is well-known and described in a number of recent, well-written reviews. The article has provided no incremental information or critical evaluations of the various, and sometimes conflicting reports in the literature, especially on the interactions between the HSCs and the cellular as well as non-cellular components of the niche. In summary, this review provides a description on the various cells and components in the bone marrow without critical evaluation of their relative roles and significance for future directions.
Author Response
Dear reviewer,
Thank you very much for your review. The article text has been thoroughly checked and linguistically revised. The revised text was edited in Track Changes mode in Word.
Sincerely,
Darina Bačenková
We have made specific changes in the following sections:
The following text has been added:
2.2.2. Leptin Receptor + Cells / Leptin + MSC osteoprogenitore
Line 430-445
LepR+ cells express and produce cytokines with an impact on hematopoiesis. In the BM, perisinusoidally localized LepR+ cells express higher levels of SCF, compared to pe-riarteriolar LepR+ cells. LepR+ cells express SCF and Cxcl12, which positively influence hematopoiesis, assisting with homing and maintenance. HSCs are influenced by SCF, which is produced by LepR+ cells and also by sinusoidal endothelial cells, while MPP is mainly influenced by SCF expressed by LepR+ cells [88].
Section 4, “Modeling the Hematopoietic Niche”, with more recent findings and advances in in vitro bone marrow simulation.
Line: 766-776
Long-term culture and maintenance of HSC/HPC is challenging and a suitable in vitro model of the BM niche is still being sought [163]. In vitro culture of HSC/HPC is still an open question and requires preservation of stemness properties. For the culture and maintenance of HSC/HPC, a specific environment of cytokines, growth factors and non-hematopoietic cell populations of heterogeneous stromal cells is essential [163]. As a medium for the culture of HSC from umbilical cord blood (UCB), Iscove's modified Dulbecco's medium (IMDM) enriched and fetal bovine and combinations of cytokines SCF, Granulocyte-macrophage colony-stimulating factor (GM-CSF), IL-3, TPO, IL-6 were mainly used. The expansion of CD34+ cells in a static culture lasting 7 days appeared to be the most suitable, thus monitoring the possible short-term liquid culture with the effect of cytokines [165].
Line: 784-800
McNiece et al. developed a multi-phase culture method with initial expansion of UCB CD34+, CD133+ and subsequent co-culture system with BM MSCs with UCB hematopoietic progenitors [170]. After isolation, CD133+ and CD34+ were cultured in a Miltenyi CliniMACS liquid culture system and culture medium with a cocktail of cytokines SCF, interleukin-3, IL-6 and G-CSF. MSCs, a population of non-hematopoietic cells of KD and other tissues were cultured adherent under standard culture conditions with the addition of fetal bovine serum in Alpha modification Minimum Essential Medium (alpha MEM) [171]. MSC KD after 70-80% monolayer were passaged and then cocultured in culture flasks for an additional 2 weeks, after which ex vivo expanded USB CD34+, CD133+ were added to the system. The coculture system was designed based on the ability of MSCs to support the proliferation of hematological progenitors by producing growth factors and adhesion molecules [172]. Overall, they achieved a 6-fold increase in CD34+ after coculture [170]. The ex vivo expansion strategy is also applicable to BM hematopoietic progenitors. The main benefit of these culture procedures is to enrich the knowledge about the BM niche and cooperating cell population and, last but not least, to increase the quality of transplantable grafts in bone marrow transplantation and also HSC/HPC after peripheral blood mobilization [173].
Line 899-909
The 3D BM niche complex forms an interface between cell populations, growth factors, cytokines and biomaterials in vitro. The 3D model simulates the bone marrow environment with a combination of a rigid component representing bone structure and a softer marrow region populated by cell populations [196]. Ravichandran tested a bone marrow adipose tissue (BMAT) model based on a combination of a GelMA scaffold with a hydrogel/medicalgrade polycaprolactone (mPCL) scaffold composite seeded with human BM stromal cells. The soft GelMa material was seeded with MSCs in combination with a tubular, macroporous mPCL scaffold that is designed for mechanical loading. The bioreactor system in which the samples were placed for three weeks in adipogenic differentiation medium provided a suitable environment during which cell viability was maintained and the cells formed lipid droplets in the cytoplasm [197].
Line 924-934
Kefallinou et al. describe the construction of an in vitro BM model, namely Bo-ne-marrow-on-a-chip (BMoC). The proposed BMoC is a scaffold-free device that com-bines microfluidic technology and co-culture of fluorescently labeled human HSPCs and MSCs. The BMoC perfusion model was constructed as a PDMS or poly(ethylene tere-phthalate) (PET) membrane in the form of a chip with the possibility of microscopic observation of cells in MSC-HSPC co-culture, which is placed in a culture dish with growth medium. Adhered MSCs on the PET membrane maintained their spin-dle-shaped morphology and were evenly distributed. In the case of PDMS membranes, MSCs did not have a homogeneous distribution and accumulated regionally. The ob-served HSPCs were homogeneously distributed. The biomimetic BM system was des-cribed as an in vitro simulation of a hematopoietic niche near the perivascular niche [200].
We have added references to the relevant parts of the text:
- Andrade-Zaldívar, H.; Santos, L.; De León Rodríguez, A. Expansion of human hematopoietic stem cells for transplantation: trends and perspectives. Cytotechnology, 2008, 56, 151-160.
- Mohamed, A.A.; Ibrahim, A.M.; El-Masry, M.W. Ex vivo expansion of stem cells: defining optimum conditions using various cytokines. Laboratory hematology: official publication of the International Society for Laboratory Hematology 2006, 12, 86-93.
- McNiece, I.K.; Robinson, S. N.; Shpall, E.J. MSC for Ex Vivo Expansion of Umbilical Cord Blood Cells. In Mesenchymal Stromal Cells: Biology and Clinical Applications, 1nd ed. New York, Springer Nature, Humana Press, New York, 2013, F4869, 485-501.
- Delorme, B.; Charbord, P. Culture and characterization of human bone marrow mesenchymal stem cells. In Tissue engineering . 2nd ed.; Totowa, New Jersey, USA, Humana Press, 2007; 140, p. 67-81.
- McNiece, I.; Harrington, J.; Turney, J. Ex vivo expansion of cord blood mononuclear cells on mesenchymal stem cells. Cytotherapy 2004, 6, 311-317.
- Purdy, M.H.; Hogan, C.J.; Hami, L. Large volume ex vivo expansion of CD34-positive hematopoietic progenitor cells for transplantation. Journal of Hematotherapy, 1995, 4, 515-525.
- Kandarakov, O., Belyavsky, A., & Semenova, E. (2022). Bone marrow niches of hematopoietic stem and progenitor cells. International journal of molecular sciences, 23(8), 4462.
- Ravichandran, A.; Meinert, C.; Bas, O. Engineering a 3D bone marrow adipose composite tissue loading model suitable for studying mechanobiological questions. Materials Science and Engineering: C 2021, 128, 112313.
- Kefallinou, D.; Grigoriou, M.; Boumpas, D.T. Mesenchymal Stem Cell and Hematopoietic Stem and Progenitor Cell Co-Culture in a Bone-Marrow-on-a-Chip Device toward the Generation and Maintenance of the Hematopoietic Niche. Bioengineering, 2024, 11, 748.

Reviewer 2 Report
Comments and Suggestions for Authors
The authors have substantially revised the text of the article, but there are a number of minor comments:
Line 41: Nestin+ MSCs are redundant when listing differentiated cells.
Line 55: mi-croenvironment, in-divid-ual
Line 71: "They have proposed" - Who are they?
Line 88: popula-tion
Line 258: "HSCs are multipotent with the ability to differentiate into HPCs, which are capable of differentiating into any blood cell type, the erythroid lineage, and the myeloid lineage. " - erythroid and myeloid are also blood cells
HSC/HPC phenotype: Look https://doi.org/10.3389/fphys.2022.1009160, it may be useful
Line 407: CXCL12 and CXCR4 regulate HSC homing in homeostasis, not only in wound healing
Reference 59 does not contain data about endosteal niche.
Lines 576-588: Please separate mesenchymal stem cells in bone marrow and multipotent mesenchymal stromal cells in culture.
Lines 590-593: What do the quotation marks mean?
Author Response
Dear reviewer,
Thank you very much for your review.
The revised text was edited in Track Changes mode in Word.
Sincerely,
Darina Bačenková
The authors have substantially revised the text of the article, but there are a number of minor comments:
Author’s response:
Thank you very much for reviewing our article.
Line 41: Nestin+ MSCs are redundant when listing differentiated cells.
- We have removed the redundant term "Nestin+ MSCs"
- Corrected sentence: “ In the BM niche HSCs, HPCs, interact with mesenchymal stem cell (MSC) populations, and differentiated cell types such as endosteal osteoblasts, leptin receptor-positive cells, CXCL12-rich reticular cells, (CAR) cells, neuroglial antigen 2 (NG2) cells [1]“.
Line 55: mi-croenvironment, in-divid-ual
- Line 55: "mi-croenvironment, in-dividual, popula-tion " errors have been fixed.
Line 71: "They have proposed" - Who are they?
- We have included the cited author in the sentence: Adams, 2006.
„Adams et al. proposed that BMM stimuli are involved in the regulation of hematopoietic stem cells through a highly specific environment.“ Adams et al. proposed that BMM stimuli are involved in the regulation of hematopoietic stem cells through a highly specific environment.
Line 258: "HSCs are multipotent with the ability to differentiate into HPCs, which are capable of differentiating into any blood cell type, the erythroid lineage, and the myeloid lineage. " - erythroid and myeloid are also blood cells
HSC/HPC phenotype: Look https://doi.org/10.3389/fphys.2022.1009160, it may be useful
- Line 258: The sentence was rewritten as recommended. „HSCs are multipotent with the ability to differentiate into HPCs, which form the basis of hematopoietic lineages through lymphoid progenitors giving rise to the B cell, T cell and NK cell population, and through myeloid progenitors, they differentiate into red blood cells, platelets, granulocytes and monocytes.“
Line 407: CXCL12 and CXCR4 regulate HSC homing in homeostasis, not only in wound healing
- The sentence was corrected and supplemented according to the recommendation:
„CXCL12 is a chemokine involved in the mobilization and homing of CD34+ HSCs in homeostasis and during wound healing processes and participate in tissue repair [53]“.
Reference 59 does not contain data about endosteal niche.
Line 330 (Reference 59)
- We have replaced the inappropriate reference with the correct one.: 59. Cordeiro‐Spinetti, E.; Taichman, R.S. The bone marrow endosteal niche: how far from the surface?. Journal of cellular biochemistry 2015, 116, 6-11.
Lines 576-588: Please separate mesenchymal stem cells in bone marrow and multipotent mesenchymal stromal cells in culture.
- We separated mesenchymal stem cells in bone marrow and multipotent mesenchymal stromal cells in culture.
New sentence added: “In humans, multipotent mesenchymal stromal cells / mesenchymal stem cells are isolated from KD aspirate, adipose tissue, and other tissues and are monolayer-cultured in vitro with a characteristic spindle-shaped shape as adherent cells [61].“
Lines 590-593: What do the quotation marks mean?
- We removed unnecessary quotation marks that were meant to emphasize the full name of the abbreviation description.
The following text has been added:
2.2.2. Leptin Receptor + Cells / Leptin + MSC osteoprogenitore
Line 430-445
LepR+ cells express and produce cytokines with an impact on hematopoiesis. In the BM, perisinusoidally localized LepR+ cells express higher levels of SCF, compared to pe-riarteriolar LepR+ cells. LepR+ cells express SCF and Cxcl12, which positively influence hematopoiesis, assisting with homing and maintenance. HSCs are influenced by SCF, which is produced by LepR+ cells and also by sinusoidal endothelial cells, while MPP is mainly influenced by SCF expressed by LepR+ cells [88].
Section 4, “Modeling the Hematopoietic Niche”, with more recent findings and advances in in vitro bone marrow simulation.
Line: 766-776
Long-term culture and maintenance of HSC/HPC is challenging and a suitable in vitro model of the BM niche is still being sought [163]. In vitro culture of HSC/HPC is still an open question and requires preservation of stemness properties. For the culture and maintenance of HSC/HPC, a specific environment of cytokines, growth factors and non-hematopoietic cell populations of heterogeneous stromal cells is essential [163]. As a medium for the culture of HSC from umbilical cord blood (UCB), Iscove's modified Dulbecco's medium (IMDM) enriched and fetal bovine and combinations of cytokines SCF, Granulocyte-macrophage colony-stimulating factor (GM-CSF), IL-3, TPO, IL-6 were mainly used. The expansion of CD34+ cells in a static culture lasting 7 days appeared to be the most suitable, thus monitoring the possible short-term liquid culture with the effect of cytokines [165].
Line: 784-800
McNiece et al. developed a multi-phase culture method with initial expansion of UCB CD34+, CD133+ and subsequent co-culture system with BM MSCs with UCB hematopoietic progenitors [170]. After isolation, CD133+ and CD34+ were cultured in a Miltenyi CliniMACS liquid culture system and culture medium with a cocktail of cytokines SCF, interleukin-3, IL-6 and G-CSF. MSCs, a population of non-hematopoietic cells of KD and other tissues were cultured adherent under standard culture conditions with the addition of fetal bovine serum in Alpha modification Minimum Essential Medium (alpha MEM) [171]. MSC KD after 70-80% monolayer were passaged and then cocultured in culture flasks for an additional 2 weeks, after which ex vivo expanded USB CD34+, CD133+ were added to the system. The coculture system was designed based on the ability of MSCs to support the proliferation of hematological progenitors by producing growth factors and adhesion molecules [172]. Overall, they achieved a 6-fold increase in CD34+ after coculture [170]. The ex vivo expansion strategy is also applicable to BM hematopoietic progenitors. The main benefit of these culture procedures is to enrich the knowledge about the BM niche and cooperating cell population and, last but not least, to increase the quality of transplantable grafts in bone marrow transplantation and also HSC/HPC after peripheral blood mobilization [173].
Line 899-909
The 3D BM niche complex forms an interface between cell populations, growth factors, cytokines and biomaterials in vitro. The 3D model simulates the bone marrow environment with a combination of a rigid component representing bone structure and a softer marrow region populated by cell populations [196]. Ravichandran tested a bone marrow adipose tissue (BMAT) model based on a combination of a GelMA scaffold with a hydrogel/medicalgrade polycaprolactone (mPCL) scaffold composite seeded with human BM stromal cells. The soft GelMa material was seeded with MSCs in combination with a tubular, macroporous mPCL scaffold that is designed for mechanical loading. The bioreactor system in which the samples were placed for three weeks in adipogenic differentiation medium provided a suitable environment during which cell viability was maintained and the cells formed lipid droplets in the cytoplasm [197].
Line 924-934
Kefallinou et al. describe the construction of an in vitro BM model, namely Bo-ne-marrow-on-a-chip (BMoC). The proposed BMoC is a scaffold-free device that com-bines microfluidic technology and co-culture of fluorescently labeled human HSPCs and MSCs. The BMoC perfusion model was constructed as a PDMS or poly(ethylene tere-phthalate) (PET) membrane in the form of a chip with the possibility of microscopic observation of cells in MSC-HSPC co-culture, which is placed in a culture dish with growth medium. Adhered MSCs on the PET membrane maintained their spin-dle-shaped morphology and were evenly distributed. In the case of PDMS membranes, MSCs did not have a homogeneous distribution and accumulated regionally. The ob-served HSPCs were homogeneously distributed. The biomimetic BM system was des-cribed as an in vitro simulation of a hematopoietic niche near the perivascular niche [200].
We have added references to the relevant parts of the text:
- Andrade-Zaldívar, H.; Santos, L.; De León Rodríguez, A. Expansion of human hematopoietic stem cells for transplantation: trends and perspectives. Cytotechnology, 2008, 56, 151-160.
- Mohamed, A.A.; Ibrahim, A.M.; El-Masry, M.W. Ex vivo expansion of stem cells: defining optimum conditions using various cytokines. Laboratory hematology: official publication of the International Society for Laboratory Hematology 2006, 12, 86-93.
- McNiece, I.K.; Robinson, S. N.; Shpall, E.J. MSC for Ex Vivo Expansion of Umbilical Cord Blood Cells. In Mesenchymal Stromal Cells: Biology and Clinical Applications, 1nd ed. New York, Springer Nature, Humana Press, New York, 2013, F4869, 485-501.
- Delorme, B.; Charbord, P. Culture and characterization of human bone marrow mesenchymal stem cells. In Tissue engineering . 2nd ed.; Totowa, New Jersey, USA, Humana Press, 2007; 140, p. 67-81.
- McNiece, I.; Harrington, J.; Turney, J. Ex vivo expansion of cord blood mononuclear cells on mesenchymal stem cells. Cytotherapy 2004, 6, 311-317.
- Purdy, M.H.; Hogan, C.J.; Hami, L. Large volume ex vivo expansion of CD34-positive hematopoietic progenitor cells for transplantation. Journal of Hematotherapy, 1995, 4, 515-525.
- Kandarakov, O., Belyavsky, A., & Semenova, E. (2022). Bone marrow niches of hematopoietic stem and progenitor cells. International journal of molecular sciences, 23(8), 4462.
- Ravichandran, A.; Meinert, C.; Bas, O. Engineering a 3D bone marrow adipose composite tissue loading model suitable for studying mechanobiological questions. Materials Science and Engineering: C 2021, 128, 112313.
- Kefallinou, D.; Grigoriou, M.; Boumpas, D.T. Mesenchymal Stem Cell and Hematopoietic Stem and Progenitor Cell Co-Culture in a Bone-Marrow-on-a-Chip Device toward the Generation and Maintenance of the Hematopoietic Niche. Bioengineering, 2024, 11, 748.

Round 3
Reviewer 1 Report
Comments and Suggestions for Authors
The major challenge is that the manuscript lacks novelty and does not provide incremental knowledge to the field. I miss a critical assessment of the cited references, and especially a critical and komparative evaluation of the various approaches to advance the field.
Author Response
Dear reviewer, thank you for your time and recommendations.
We have thoroughly revised the manuscript, with improved language, a shortened and focused conclusion, and a stronger emphasis on in vitro modeling of the hematopoietic niche including a summary table of current models and future outlook.
Best regards,
Bačenková